# Elevated Liver Fibrosis Progression in Isolated PSC Patients and Increased Malignancy Risk in a PSC-IBD Cohort: A Retrospective Study

**DOI:** 10.3390/ijms242015431

**Published:** 2023-10-21

**Authors:** Florian Rennebaum, Claudia Demmig, Hartmut H. Schmidt, Richard Vollenberg, Phil-Robin Tepasse, Jonel Trebicka, Wenyi Gu, Hansjoerg Ullerich, Iyad Kabar, Friederike Cordes

**Affiliations:** 1Department of Internal Medicine B, Gastroenterology, Hepatology, Endocrinology and Clinical Infectiology, University Hospital Münster, 48149 Münster, Germany; claudia.n.demmig@gmx.de (C.D.); richard.vollenberg@ukmuenster.de (R.V.); phil-robin.tepasse@ukmuenster.de (P.-R.T.); jonel.trebicka@ukmuenster.de (J.T.); wenyi.gu@ukmuenster.de (W.G.); hansjoerg.ullerich@ukmuenster.de (H.U.); 2Department of Hepatology, Gastroenterology and Transplantation Medicine, University Hospital Essen, 45147 Essen, Germany; hartmut.schmidt@uk-essen.de; 3Department of Internal Medicine, University Teaching Hospital Raphaelsklinik Münster, 48143 Münster, Germany; i.kabar@alexianer.de; 4Department of Internal Medicine II Gastroenterology, University Teaching Hospital Euregio-Klinik Nordhorn, 48527 Nordhorn, Germany; friederike.cordes@euregio-klinik.de

**Keywords:** dominant stenosis, PSC, primary sclerosing cholangitis, CRC, colorectal cancer, IBD, inflammatory bowel disease, OLT, orthotopic liver transplantation

## Abstract

Primary sclerosing cholangitis (PSC) is a chronic cholestatic liver disease often associated with inflammatory bowel disease (IBD), particularly ulcerative colitis (CU), and rarely with Crohn’s disease (CD). Various long-term analyses show different rates of cancer and the need for orthotopic liver transplantation (OLT) in patients with isolated PSC and with concomitant IBD, respectively. However, data on the detailed course of PSC with or without IBD are limited. We aimed to analyze the clinical disease course of PSC patients without IBD compared to PSC patients with UC and CD, respectively. A retrospective data analysis of patients with isolated PSC (n = 41) and of patients with concomitant IBD (n = 115) was performed. In detail, PSC disease characteristics including occurrence of dominant stenoses, liver cirrhosis, OLT and malignancy, as well as the temporal course of PSC activity and disease progression, were analyzed. A multivariable Cox regression model and a Fine–Gray competing risk model were further used for the independent risk factor analysis of cirrhosis development and OLT. Patients with isolated PSC were significantly older at first diagnosis than patients with PSC-IBD (39 vs. 28 years, *p* = 0.02). A detailed analysis of the course of PSC revealed a faster PSC progression after initial diagnosis in isolated PSC patients compared to PSC-IBD including significantly earlier diagnosis of dominant stenoses (29 vs. 74 months, *p* = 0.021) and faster progression to liver cirrhosis (38 vs. 103 months, *p* = 0.027). Patients with isolated PSC have a higher risk of developing cirrhosis than patients with PSC-IBD (Gray’s test *p* = 0.03). OLT was more frequently performed in male patients with isolated PSC compared to males with coincident IBD (48% (n = 13) vs. 33% (n = 25), *p* = 0.003). Colorectal carcinoma was significantly more often diagnosed in patients with PSC-IBD than in isolated PSC (8.7% vs. 0%, *p* = 0.042). Patients with isolated PSC seem to have a different clinical course of disease than PSC patients with concomitant IBD characterized by a more pro-fibrotic disease course with earlier onset of liver cirrhosis and dominant stenosis but with less malignancy. These data may be interpreted as either a more progressive disease course of isolated PSC or a later diagnosis of the disease at an advanced disease stage. The different clinical courses of PSC and the underlying mechanisms of the gut–liver axis need further attention.

## 1. Introduction

Primary sclerosing cholangitis (PSC) is a chronic, slowly progressive cholestatic liver disease that affects the biliary duct system, causes chronic bile destruction and leads to biliary cirrhosis and portal hypertension [1,2]. Clinical presentation results in recurrent cholangitis, cholestasis, fatigue and jaundice [3]. Of note is an increased risk of malignant diseases, especially hepatobiliary carcinoma, but also colorectal carcinoma, which is associated with a fourfold increased overall mortality compared to the general population [4,5]. The median survival time after initial diagnosis of PSC without receiving liver transplantation varies between 10 and 21 years [4,6,7,8]. So far, orthotopic liver transplantation (OLT) is the only curative therapy and treatment of choice for end-stage liver disease [9]. The etiology of PSC has not been adequately clarified. Some studies point to an immunological cause characterized by the infiltration of lymphocytes into the portal tract, leading to bile duct destruction, which is caused by the production of inflammatory cytokines [10,11]. Currently, an interaction between genetic predisposition, environmental factors and dysregulated immune processes is assumed to lead to the disease pattern [12]. In this context, PSC is strongly associated with IBD. Patients with PSC have coincident ulcerative colitis (UC) in more than 80% of cases, men are more often affected than women and seem to have an increased risk of OLT and malignancy [4,7]. The median age at onset of PSC ranges from 20 to 40 years [4]. Interestingly, in patients with concomitant IBD, PSC is more frequently diagnosed after initial IBD diagnosis, which is explained by established surveillance strategies [13]. The variable and subclinical course of PSC with fluctuating elevated cholestasis parameters possibly is associated with the difficulty of assessing the diagnosis and stage of the disease, especially in patients without coexisting IBD [12]. In this context, it is noteworthy that Lunder and coauthors detected a threefold higher prevalence of subclinical PSC in IBD patients by using screening magnetic resonance cholangiography (MRC) in IBD patients [14]. The results of these studies support the subclinical, and at the same time progressive, course of the disease.

PSC-IBD might represent a unique disease phenotype that is characterized by a distinct course of disease. This is revealed by recent studies indicating a worse prognosis of PSC in association with IBD, especially UC, which is highlighted by an increased risk of colorectal carcinoma [15,16,17,18]. PSC without coexisting IBD is rare overall and few data exist on disease progression and clinical course. To our knowledge, data are lacking in analyzing the detailed clinical course of PSC in patients with or without IBD.

The aim of this study was the assessment of the temporal disease course, including laboratory parameters, disease activity indices and episodes of cholangitis in patients with isolated PSC compared to patients with PSC and concomitant IBD. Furthermore, biliary complications such as biliary obstructions/dominant stenosis, need for OLT, cirrhosis and malignancy were assessed.

## 2. Results

### 2.1. Demographics

Our retrospective analysis identified 156 patients with PSC, including 41 patients with isolated PSC (26%) and 115 patients (74%) with concomitant IBD (Table 1, Appendix A). The median age at PSC diagnosis was significantly higher in patients with isolated PSC compared to patients with coexistent IBD (39 years (30–51 years) vs. 28 years (22–46 years), *p* = 0.02). The median age at first diagnosis of IBD in UC was 27 years and in CD 21 years. IBD was diagnosed earlier than PSC in 93% (n = 14) of the CD cohort and in 83% (n = 77) of the UC cohort. Large-duct PSC was the predominant PSC subtype in both groups, with an occurrence of 88% (n = 36) in PSC patients with isolated IBD and 95% (n = 109) in patients with PSC-IBD.

### 2.2. Liver Cirrhosis and Dominant Stenoses

Dominant stenoses of the bile ducts are associated with recurrent episodes of cholangitis and reduced transplant-free survival [19,20].

In our study, a significantly shorter time interval between the initial diagnosis of PSC and the first diagnosis of dominant stenoses was demonstrated in isolated PSC compared to PSC-IBD (29 months (4–96 months) vs. 74 months (43–130 months), *p* = 0.021; Figure 1A). Interestingly, we detected significant differences in the occurrence of dominant stenoses between patients with isolated PSC and patients with coexisting IBD. Especially in the first five years after the initial diagnosis of PSC, a significantly higher rate of dominant stenoses was diagnosed in isolated PSC patients compared to PSC-IBD (68% (n = 15) vs. 39% (n = 16), *p* = 0.003; Figure 1B). Furthermore, overall rates of cholangitis were significantly higher in patients with isolated PSC (cholangitis: 76% (n = 48) vs. 59% (n = 58), *p* = 0.014; Table 2). Comparable results were obtained by examining the time intervals between the first diagnosis of PSC and the first detection of liver cirrhosis. In the isolated PSC cohort, the time between the first diagnosis of PSC and the detection of liver cirrhosis was significantly shorter than in the PSC-IBD group (38 months (0–104 months) vs. 103 months (32–206 months), *p* = 0.027, Figure 1C). The diagnosis of liver cirrhosis was made significantly more frequently in isolated PSC within the first five years after initial diagnosis compared to PSC-IBD (57% (n = 8) vs. 25% (n = 11), *p <* 0.001; Figure 1D). Using the Fine and Gray competing risk model (OLT as competing risk) and Cox regression model, it was demonstrated that patients with isolated PSC have a higher risk of developing cirrhosis than patients with PSC-IBD (Gray’s test *p* = 0.03; Figure 2). No significant differences concerning dominant stenoses, cholangitis and liver cirrhosis were detected in the subgroup analyses between isolated PSC and PSC-UC and PSC-CD, respectively. Taken together, our results show a shorter interval for the manifestation of PSC-associated complications in patients with isolated PSC compared to PSC-IBD patients.

### 2.3. Change of PSC Disease Activity over Time

Prognostic scores were evaluated to assess the clinical courses of PSC and are presented in Table 2. Using the Mayo Risk Score as a parameter of PSC disease progression [21], a significant higher overall score was found in patients with isolated PSC compared to patients with coexisting IBD (2.46 (1.22–3.56) vs. 1.05 (0–2.28), *p* = 0.015; Table 2). The Amsterdam-Oxford PSC score, Child-Pugh-Score and MELD Score, however, showed no significant differences between the two cohorts (Amsterdam-Oxford-PSC Score: 2.27 (3.11–1.99) vs. 2.16 (1.66–2.97), *p* = 0.401; Child-Pugh Score: 7 (5–8) vs. 7 (5–8), *p* = 0.256; Meld-Score: 13 (7–16) vs. 12 (7–16), *p* = 0.873; Table 2). Laboratory parameters between the respective PSC subgroups were examined over the entire observation period (Table 3). During the first five years after diagnosis, bilirubin and AP revealed no significant differences, while yGT, aspartate aminotransferase (AST) and alanine aminotransferase (ALT) revealed significantly higher serum levels in patients with isolated PSC compared to PSC-IBD (Figure 3A–F).

### 2.4. Liver Transplantation

OLT is currently the only curative treatment for PSC [22]. OLT was more frequently needed in patients with isolated PSC compared to PSC patients with concomitant IBD, however, these data did not reach statistical significance (36% (n = 15) vs. 26% (n = 30), *p* = 0.14; Table 1). Interestingly, an overall analysis of all included patients revealed a significantly higher transplantation rate in men compared to women (37% (n = 38) vs. 13% (n = 7), *p* = 0.002). Further subgroup analyses demonstrated a significantly increased rate of transplantation in males with isolated PSC compared to male PSC-IBD patients (48% (n = 13) vs. 33% (n = 25), *p* = 0.003; Table 1). Both cohorts were normally distributed in terms of gender. The time intervals between the initial diagnosis of PSC and liver transplantation did not show any relevant differences between the two cohorts.

Next, we performed logistic regression analyses to identify risk factors for the need for OLT. As expected, liver cirrhosis was demonstrated to be an independent and strong risk factor that highly increased the risk for liver transplantation (OR 16.9, 95% CI 4.53–56.2; *p* < 0.001; Figure 4). Furthermore, analogous to an increased occurrence of liver transplantation in male patients, the female gender revealed a protective factor that significantly reduced the risk for transplantation (OR 0.16, 95% CI 0.04–0.70; *p* = 0.015). Interestingly, concomitant IBD also represented a protective factor with reduced risk of OLT (OR 0.23, 95% CI 0.06–0.88, *p* = 0.031; Figure 4).

### 2.5. Neoplasia

In addition to biliary complications, PSC with concomitant IBD is associated with an increased rate of hepatobiliary and colorectal carcinoma [23]. Thus, we next aimed to analyze the occurrence of malignancies in our cohorts. Interestingly, no colorectal cancer (CRC) was detected in patients with isolated PSC compared to a CRC rate of 8.4% in PSC-IBD patients (0% (n = 0) vs. 8.4% (n = 10), *p* = 0.043); Figure 5A,B). Subgroup analysis confirmed a significantly higher incidence of CRC in UC patients compared to isolated PSC (0% (n = 0) vs. 9% (n = 9), *p* = 0.041; Table 4), while no significant differences were detected in PSC-CD patients compared to isolated PSC (7% (n = 1), *p* = 0.095). No relevant differences between isolated PSC and PSC-IBD were found with regard to the diagnosis of hepatobiliary carcinomas (12% (n = 5) vs. 9.6% (n = 11), *p* = 0.415; Table 4). In subgroup analyses of isolated PSC and PSC patients with concomitant UC and CD, respectively, no relevant differences were found concerning hepatobiliary carcinoma. However, it is worth noting that no hepatobiliary carcinoma was detected in the PSC-CD group, while 11% (n = 11) were found in PSC-UC patients. As demonstrated in Figure 1, dominant stenoses, which are discussed as premalignant lesions with an increased risk for hepatobiliary carcinoma [24], were significantly more frequently found in isolated PSC patients compared to PSC-IBD patients (58% (n = 22) vs. 40% (n = 41); *p* = 0.042). No significant differences were found in subgroup analyses concerning PSC-UC and PSC-CD patients compared to isolated PSC.

## 3. Discussion

In our retrospective data analysis, we examined the clinical disease course of PSC in patients with and without concomitant IBD. We can demonstrate that patients with isolated PSC revealed a significant higher disease activity, including higher rates and an earlier occurrence of dominant stenoses; progressed development of liver cirrhosis; higher levels of liver parameter during the first 5 years of the disease course; and a later age of PSC onset compared to patients with concomitant IBD. These data demonstrate a profibrotic course of disease in patients with isolated PSC, which is characterized by advanced disease stage and a stronger disease progression within the first years after diagnosis.

PSC is a heterogeneous disease that, in addition to the frequent association with IBD, has different clinical subphenotypes with a diverse range of clinical manifestations [25]. Regardless of the phenotype of PSC, the disease leads to the development of liver cirrhosis over a period of 10–12 years due to chronic inflammation of the bile duct system and recurrent cholangitis [6,26,27]. There is evidence that PSC has a different course of disease dependent on the coincidental diagnosis of IBD. For example, a meta-analysis demonstrated a bimodal distribution of the age of onset of PSC [28]. Specifically, a study from Japan showed that patients diagnosed with PSC at an older age had lower rates of IBD [29]. Similar results were shown in a North American study, where the rate of IBD was significantly lower in older PSC patients [30]. In addition, an Australian multicenter study found a significantly lower rate of liver cirrhosis in patients with PSC and co-occurring IBD compared to patients with isolated PSC [31]. In accordance with these data, we can demonstrate in our study that patients with isolated PSC are diagnosed at a significantly later age compared to PSC patients with concomitant IBD. These findings coincide with higher manifestation rates of dominant stenosis and liver cirrhosis during the first five years after PSC onset and a higher risk of developing cirrhosis in isolated PSC. On the one hand, our data hint at a delayed diagnosis of PSC in patients without IBD, which leads to a progressed disease at first presentation and an earlier manifestation of disease-associated complications like liver cirrhosis and dominant stenoses. This hypothesis is supported by the overall difficulty in diagnosing PSC [32,33]. The challenge of diagnosing PSC is due to the often variable and non-specific symptoms of PSC, such as fatigue or pruritus, which delay diagnosis and clarification by years [34]. For instance, patients diagnosed with IBD and elevated cholestasis parameters are more frequently evaluated for PSC by MRCP or ERC, allowing for more timely diagnosis. This may serve as a possible reason for our observations of the shortened time interval between the first diagnosis and the detection of liver cirrhosis or dominant stenoses in PSC patients without concomitant IBD.

The hypothesis of distinct characteristics of isolated PSC is further underlined by epidemiological studies from Asia. In this context, Hirano et al. demonstrated a lower rate of IBD in PSC patients at older ages [35]. A study from the USA also showed an age-dependent incidence of IBD in PSC patients, which is associated with low rates of IBD at older ages [36]. The similar data of the late age of onset of PSC with low coincidence of IBD in different ethnic populations suggest a distinct subtype of PSC and do not appear to be limited to specific population groups [35]. The higher rate of liver cirrhosis, dominant stenoses in the first five years, and risk for developing liver cirrhosis may also indicate a more profibrotic course and severe disease progression. A paper by Angulo and coauthors examined the course of PSC using histological stages over a period of five years. After five years, the patient cohort showed a significant increase concerning liver cirrhosis: from 44% to 93% in PSC patients [37]. These results underline the hypothesis of an overall more progressive course in patients with isolated PSC compared to patients with PSC-IBD in the first five years after diagnosis.

Taking these studies and our own data into account, a certain PSC phenotype that seems to be associated with older age at onset and faster disease progression, characterized by higher rates of liver cirrhosis and dominant stenoses in the first five years after diagnosis and less concurrent IBD, can be speculated. Additionally, missing surveillance strategies in PSC patients with missing IBD possibly overlap the distinct disease course of isolated PSC, leading to a delay in diagnosis and the worsening of disease progression.

The progressive course of PSC leads to recurrent bile duct inflammation over a period of 10–12 years with the development of liver cirrhosis [6,38]. With increasing disease progression, a large proportion of PSC patients must undergo OLT [9]. Depending on the study population, the rate of OLT in PSC patients varies between 14.6% and 44%, with males tending to be transplanted more frequently [22,39]. Based on our data, using competing risk analysis, a higher risk of developing cirrhosis was demonstrated in isolated PSC patients even when we used liver transplantation as a competing risk for developing cirrhosis. In addition to liver cirrhosis, we were able to identify the male gender as a possible risk factor for liver transplantation. Similar results were shown in a large multicenter retrospective data collection effort undertaken by Weismueller and coauthors, which demonstrated an increased risk for OLT in men.

Additionally, we could demonstrate that OLT was significantly more frequently needed in males with isolated PSC compared to male PSC patients with concomitant IBD. A possible explanation for higher rates of OLT in isolated PSC may be the significantly later time of initial PSC diagnosis. Furthermore, there may already be other comorbidities such as chronic renal insufficiency, which are included in the organ allocation by the MELD score [40]. Another explanation for the greater need for transplantation in patients without co-occurring IBD may be the significantly higher rate of dominant stenoses and episodes of cholangitis. Dominant stenoses and recurrent cholangitis as a measure of increased disease activity of PSC are taken into account with standard exception (SE) points in organ allocation in Germany, leading to a shorter waiting time for a liver transplant [41].

Several prospective cohort studies have shown an increased incidence of gastrointestinal carcinomas associated with PSC [42]. In detail, PSC was shown to be an independent risk factor for the development of colorectal cancer in patients with ulcerative colitis [43,44]. Concerning the development of malignancy, our data analysis showed a significant difference in the incidence of colorectal cancer, which occurred exclusively in patients with PSC-IBD. The results may be explained by colonoscopy screening and exhaustive surveillance, which is recommended by various studies and guidelines, revealing a higher incidence of CRC in patients with PSC-UC [45,46,47]. On the other hand, similar results have been achieved in a large meta-analysis showing a fourfold increased risk of colorectal carcinoma in patients with PSC and coincidental IBD compared to patients with UC alone [48]. Our study provides evidence that PSC increases the carcinogenesis of colorectal cancers in patients with co-occurring IBD. The absence of colorectal carcinomas in PSC patients without coexisting IBD highlights a specific PSC subtype with different mechanisms of carcinogenesis.

Chronic inflammation of the bile ducts leads to hepatobiliary carcinoma in 10 to 20% of PSC patients during the course of the disease, which is a major cause of increased mortality in PSC [49,50,51]. The investigation of hepatobiliary carcinomas did not reveal any significant differences between isolated PSC and PSC-IBD patients. Observing the group of isolated PSC, we detected a significantly higher incidence of dominant stenoses in comparison to PSC-IBD patients. It is suggested that the presence of dominant stenoses is associated with a worse prognosis of PSC due to the risk of developing carcinoma of the bile duct [52,53,54]. The more frequent occurrence of dominant stenoses in the cohort of isolated PSC patients may explain the significantly higher rate of bacterial cholangitis, which is explained by impaired bile flow due to biliary strictures, leading to the chronic deterioration of liver function [55,56]. Dominant stenoses were associated with a significantly increased risk of developing hepatobiliary carcinoma in a 25-year follow-up study and is considered a premalignant condition [24,57]. Regarding our own data, although we could not find a higher incidence of hepatobiliary carcinoma in isolated PSC patients, possibly due to a limited follow up interval [24], we note a higher rate of dominant stenoses, which is discussed as a premalignant leasing and hints at a distinct risk for cholangiocarcinoma development in isolated PSC patients.

Limitations of our study include the retrospective monocentric design, with its lack of randomization. As a further limitation, immunosuppressive drugs in the IBD group may have influenced the results of the laboratory parameters and biased the statistical data. Furthermore, the course of PSC in patients with IBD may have been positively influenced by the use of various immunosuppressants. However, this design enabled a detailed analysis of the disease course with the opportunity to further characterize the course of PSC, which has not been performed so far to our knowledge.

## 4. Materials and Methods

### 4.1. Patients

We performed a retrospective single-center study analyzing 156 patients with PSC during 2005–2020. In detail, patients with PSC and concomitant IBD (defined as PSC-IBD, n = 115) were compared to patients with isolated PSC (n = 41). Patients with PSC-IBD were subclassified into patients with PSC and UC (n = 100) and Crohn’s disease (CD; n = 15) (Appendix A). PSC was subclassified in large-duct PSC (n = 145) and small-duct PSC (n = 11) All patients’ medical charts were reviewed, and patients were excluded if secondary causes of sclerosing cholangitis were existing. Patients with IgG-4 cholangiopathy were excluded. The endpoints were OLT and death. PSC diagnosis, as well as the diagnosis of bile duct strictures such as dominant stenosis, included typical radiologic findings of the intrahepatic and extrahepatic bile duct system using endoscopic retrograde cholangioscopy (ERC), direct spy-glass-cholangioscopy, MRC or percutane transhepatic cholangiography (PTC). Intestinal bowel disease was defined by different upper and lower endoscopic findings, such as ulcerative inflammation of the colon, fissures, intestinal stenosis and fistula and histopathologic results. Criteria for PSC activity were laboratory tests with at least twice-elevated cholestatic parameters using bilirubin, gamma-glutamyl transpeptidase (yGT), serum alkaline phosphatase (AP) and serum aminotransferases [58]. Furthermore, histopathological records of the liver were analyzed to define small-duct PSC, which typically has no bile duct destructions in the cholangiography [59].

### 4.2. Follow-Up Analysis of the PSC Disease Course

All patients’ medical charts were reviewed concerning the existence of liver cirrhosis, OLT, malignancy such as hepatobiliary carcinoma, colorectal carcinoma and high-grade intraepithelial neoplasia, death, laboratory tests, biliary endoscopic findings (extra and intrahepatic lesions such as dominant stenoses), the existence of esophageal varices and episodes of cholangitis with the need for antibiotic therapy. The diagnosis of liver cirrhosis was based on histological analysis, including the staging of fibrosis and the grading of inflammation using scoring systems developed by Batts and Ludwig and Desmet, as well as typical ultrasound findings [60,61,62,63].

The definition of cholangitis included the combination of fever with a temperature at least >38.3°, abdominal pain in the right upper quadrant and elevated bilirubin from baseline. Episodes of post-ERC cholangitis were excluded. Dominant stenosis was defined as biliary stenosis < 1.5 mm in the common bile duct or <1 mm in the hepatic bile duct within 2 cm of the bifurcation [20,64]. All patients received ursodeoxycholic acid in a dosage of 15–30 mg/kg body weight.

Additionally, specific prognostic scores for PSC activity such as the Mayo-Risk-Score, containing the parameters age, bilirubin, albumin and serum aminotransferases, and the Amsterdam-Oxford PSC score, including the parameters age, subtype of PSC, bilirubin, alkaline phosphatase, albumin, serum aminotransferases and platelets, were assessed [65]. Furthermore, prognostic scores to predict survival at end-stage liver disease, including the Meld Score and the Child-Pugh-Score were carried out [66]. The endpoints in this study included death and liver transplantation.

### 4.3. Statistical Analyses

Statistical analyses were carried out using IBM SPSS Statistics 25.0 (IBM Corporation, Armonk, NY, USA). The Chi-Square test with continuity correction or Fisher’s exact test was used to analyze categorical data. Continuous data were compared using the non-parametric Mann–Whitney U-test. A statistical significance was assumed if *p* < 0.05.

Univariable and multivariable binary logistic regression was performed to determine risk factors for OLT. Furthermore, we compared the group of patients with isolated PSC diagnosis and the patients with PSC-IBD for the development of cirrhosis using the Fine and Gray competing risk model (OLT as competing risk) with SAS OnDemand for Academics. The Cox regression model was used to investigate the independent risk factors for cirrhosis development. All the potential variables at baseline that were significant (*p* < 0.05) in the univariable analysis or reported in previous studies were included in the multivariable analysis.

## 5. Conclusions

In summary, our retrospective data analysis of PSC patients without coexisting IBD displays a different clinical course compared to patients with PSC-IBD. It is characterized by a profibrotic course of PSC, including a generally higher rate of dominant stenoses and a shorter time interval from the initial diagnosis to cirrhosis or dominant stenoses and a higher risk for the development of liver cirrhosis. PSC patients with dominant stenoses and older age should be evaluated early for possible OLT and require appropriate screening programs to detect hepatobiliary carcinomas. Further prospective data analysis with a large cohort is required to confirm our results and to detect the underlying pathomechanisms of the gut–liver axis.

## Figures and Tables

**Figure 1 ijms-24-15431-f001:**
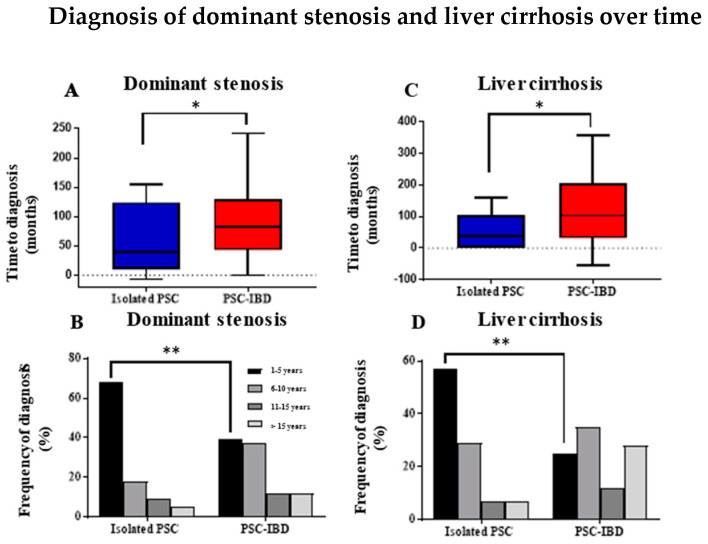
Median time interval between diagnosis of PSC and dominant stenosis (**A**) and liver cirrhosis (**C**). (**A**) indicates the median time interval in months between diagnosis of PSC and dominant stenosis (Isolated PSC 29 months vs. PSC-IBD 74 months, *p* < 0.02). (**C**) shows the median time interval between diagnosis of PSC and liver cirrhosis (Isolated PSC 38 months vs. PSC-IBD 103 months, *p* < 0.027). (**B**,**D**) reveals detection of dominant stenoses and liver cirrhosis in relation to the follow-up time interval. The diagnosis of dominant stenoses was significantly increased in the first five years after diagnosis of PSC (68% (n = 15) vs. 39% (n = 16), *p* = 0.003, (**B**)). (**D**) illustrates a significantly higher incidence of liver cirrhosis in the first five years after initial diagnosis of PSC (57% (n = 8) vs. 25% (n = 11), *p* < 0.001). * *p* < 0.05, ** *p* ≤ 0.01. PSC, primary sclerosing cholangitis; IBD, inflammatory bowel disease.

**Figure 2 ijms-24-15431-f002:**
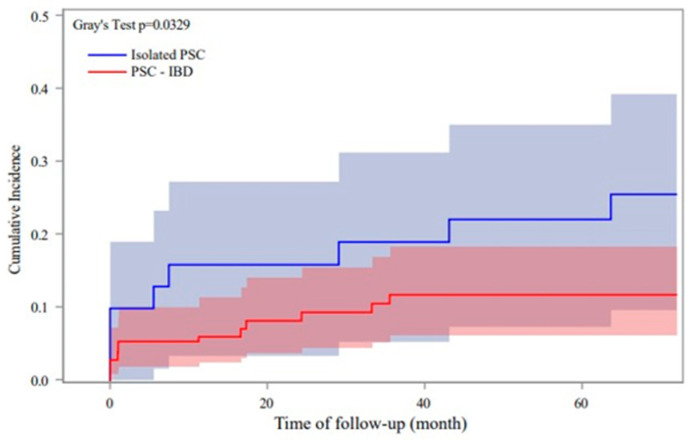
The figure shows the evolution of liver cirrhosis between isolated PSC patients and PSC-IBD using Fine and Gray competing risk model (liver transplantation as competing risk). Cox regression model was used to investigate the independent risk factors for cirrhosis development. Patients with isolated PSC have a significantly higher risk of developing cirrhosis over the observation period (Gray’s test *p* = 0.03). PSC, primary sclerosing cholangitis; IBD, inflammatory bowel disease.

**Figure 3 ijms-24-15431-f003:**
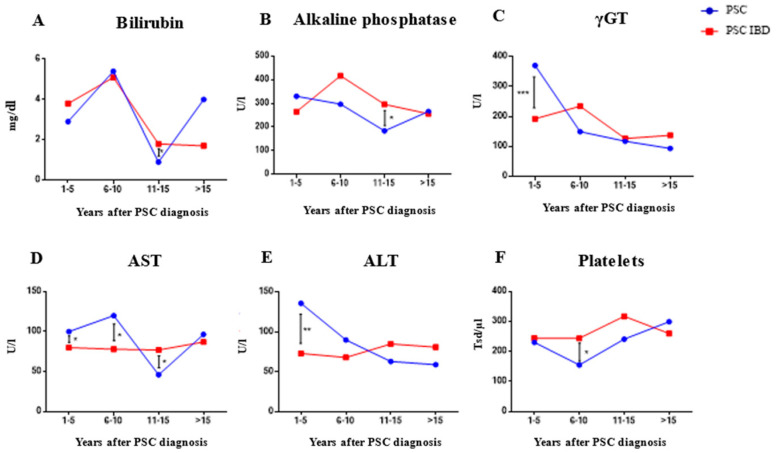
15-year follow-up of laboratory parameters between PSC-IBD and isolated PSC patients. (Fisher Exact Test; **: p* < 0.05, ** *p* ≤ 0.01, *** *p* ≤ 0.001). AST, aspartate aminotransferase; ALT, alanine aminotransferase; yGT, gamma-glutamyl transpeptidase; PSC, primary sclerosing cholangitis; IBD, inflammatory bowel disease.

**Figure 4 ijms-24-15431-f004:**
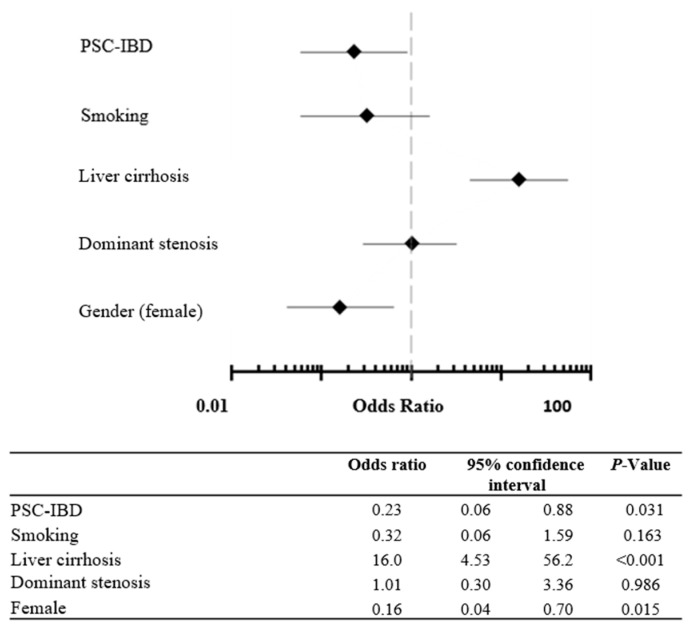
Applying logistic regression to calculate odds ratios with 95% confidence intervals identified liver cirrhosis (*p* < 0.001) and male gender as risk factors (*p* = 0.015) for liver transplantation (Figure A). PSC-IBD was identified as a protective factor for liver transplantation (*p* = 0.031). PSC, primary sclerosing cholangitis; IBD, inflammatory bowel disease.

**Figure 5 ijms-24-15431-f005:**
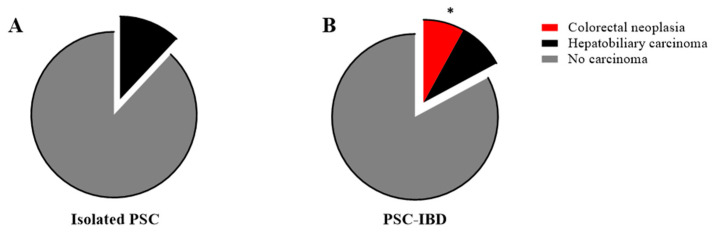
Figure (**A**,**B**) illustrate the incidence of colorectal neoplasia and hepatobiliary carcinoma between isolated PSC and PSC-IBD patients in percentage. In the isolated PSC cohort, no colorectal neoplasia was detected. (Isolated PSC 0% (n = 0) vs. PSC-IBD 8.4% (n = 10), Fisher Exact Test; *: *p* = 0.042) No significance between both groups concerning hepatobiliary carcinoma (Isolated PSC 12% (n = 5) vs. PSC-IBD 9.6% (n = 11), Fisher Exact Test; *p* = 0.415). * *p* < 0.05. PSC, primary sclerosing cholangitis; IBD, inflammatory bowel disease.

**Table 1 ijms-24-15431-t001:** Demographics and basic characteristics of patients with isolated PSC, PSC-IBD and subgroups PSC-UC and PSC-CD.

	Isolated PSC	PSC-IBD	IBD-Subgroups				
			*p*-Value	UC	*p*-Value	CD	*p*-Value
Basic Characteristics							
Number of patients [n (%)]	41 (26)	115 (74)		100 (64)		15 (10)	
Age [median; IQR]	53 (39–60)	48 (36–57)	0.145	49 (37–57)	0.199	37 (29–57)	0.141
Gender [male%]	27 (66%)	76 (66%)	0.562	67 (67%)	0.522	9 (60%)	0.459
Age at diagnosis PSC [median; (IQR)]	39 (30–51)	28 (22–46)	0.02	33 (22–44)	0.018	28 (18–52)	0.291
Age at diagnosis IBD [median; (IQR)]	[x]	24 (17–38)	[x]	27 (17–39)	[x]	21 (18–31)	[x]
Follow-up after diagnosis PSC [months; (IQR)]	108 (34–214)	107 (46–194)	0.823	112 (47–213)	0.92	78 (33–121)	0.162
Subtype of PSC							
Large-duct PSC [n %]	36 (88%)	109 (95%)	0.128	96 (96%)	0.081	13 (87%)	0.612
Small-duct PSC [n %]	5 (12%)	6 (5%)	0.128	4 (4%)	0.081	2 (13%)	0.612
OLT [n %]	15 (36%)	30 (26%)	0.142	27 (27%)	0.21	3 (20%)	0.199
OLT in male [n %]	13 (48%)	25 (33%)	0.003	22 (33%)		3 (33%)	
Duration form diagnosis of PSC to LT [months; (IQR)]	110 (7–173)	87 (16–191)	0.673	95 (16–196)	0.52	35 (7–35)	0.373
Death [n %]	2 (5%)	11 (10%)	0.516	10 (10%)	0.506	1 (7%)	0.615

PSC = primary sclerosing cholangitis: UC = ulcerative colitis; CD = Crohn’s disease, IBD = intestinal bowl disease, OLT = orthotopic liver transplantation, IQR = interquartile range; [x] = not examined.

**Table 2 ijms-24-15431-t002:** Demographics, endoscopic findings and clinical scores of patients with isolated PSC, PSC-IBD and subgroups PSC-UC and PSC-CD.

	Isolated PSC	PSC-IBD	IBD-Subgroups				
			*p*-Value	UC	*p*-Value	CD	*p*-Value
Endoscopic and general findings							
Liver cirrhosis [n (%)]	14 (41%)	43 (43%)	0.508	39 (44%)	0.458	4 (33%)	0.452
Time diagnosis PSC to liver cirrhosis [months; (IQR)]	38 (0–104)	103 (32–206)	0.027	103 (32–207)	0.017	59 (9–103)	0.959
Dominant stenosis [n (%)]	22 (58%)	41 (40%)	0.042	36 (41%)	0.059	5 (33%)	0.095
Time Diagnosis PSC to DS [months; (IQR)]	29 (4–96)	74 (43–130)	0.021	85 (45–150)	0.016	63 (17–101)	0.564
Varices upper GI [n (%)]	20 (55%)	27 (42%)	0.126	24 (43%)	0.165	3 (33%)	0.207
Time Diagnosis PSC to varices [months (IQR)]	63 (14–138)	88 (21–199)	0.189	97 (22–214)	0.137	60 (9)	0.898
Clinical characteristics							
Episodes of cholangitis [n (%)]	48 (76%)	85 (59%)	0.014	73 (59%)	0.014	12 (63%)	0.202
Need for antibiotics [n (%)]	46 (100%)	88 (88%)	0.009	76 (87%)	0.007	12 (92%)	0.22
Clinical Scores							
Amsterdam-PSC-Score [median (IQR)]	2.27 (3.11–1.99)	2.16 (1.66–2.97)	0.401	2.27 (1.66–3.05)	0.617	1.84 (1.03–2.06)	0.056
Mayo-Risk-Score [median (IQR)]	2.46 (1.22–3.56)	1.05 (0–2.28)	0.015	1.62 (0–2.28)	0.024	0.9 (0.37–1.24)	0.048
Meld-Score [median (IQR)]	13 (7–16)	12 (7–16)	0.873	11 (7–16)	0.779	14 (10–18)	0.067
Child-Pugh-Score [median (IQR)]	7 (5–8)	7 (5–8)	0.256	7 (5–8)	0.255	7 (6–7)	0.577

UC = ulcerative colitis; CD = Crohn‘s disease, IBD = intestinal bowel disease, OLT = orthotopic liver transplantation, IQR = interquartile range.

**Table 3 ijms-24-15431-t003:** Laboratory parameters in active disease of patients with isolated PSC, PSC-IBD and subgroups PSC-UC and PSC-CD.

Laboratory Parameters [Median (IQR)]	Isolated PSC	PSC-IBD	IBD-Subgroups				
			*p*-Value	UC	*p*-Value	CD	*p*-Value
Bilirubin (mg/dL)	3.4 (1.6–5.8)	3.7 (1.2–7.5)	0.078	3.3 (1.2–6.9)	0.972	5.9 (1.9–11.1)	0.129
Gamma-glutamyl transpeptidase (U/I)	229 (230–450)	182 (83–352)	0.042	184 (86–363)	0.076	138 (67–282)	0.044
Serum alkaline phosphatase (U/I)	291 (192–396)	300 (196–526)	0.267	299 (196–522)	0.335	366 (194–534)	0.277
Aspartate aminotransferase (U/I)	96 (68–135)	78 (54–123)	0.042	75 (49–111)	0.018	100 (70–142)	0.861
Alanine aminotransferase (U/I)	100 (60–214)	74 (48–129)	0.039	74 (46–125)	0.023	82 (55–208)	0.817
Albumin (g/dL)	3.5 (2.7–4.1)	3.8 (3.2–4.4)	0.154	3.70 (3.1–4.4)	0.335	4.2 (3.4–4.7)	0.214
Hemoglobin (g/dL)	12.8 (11–13.8)	13 (11.2–13.0)	0.734	12.9 (11.1–14.0)	0.881	13.3 (11.6–14.1)	0.364
Platelets (Tsd./µL)	225 (150–292)	258 (151–374)	0.074	260 (155–387)	0.035	181 (127–289)	0.18
Leukocytes (Tsd./µL)	7.7 (6.1–10.0)	7.4 (4.9–11.0)	0.499	7.4 (5.1–11.4)	0.748	6.6 (4.3–8.3)	0.084
Lymphocytes (Tsd./µL)	2.0 (1.9–2.0)	1.7 (1.5–2.4)	0.149	1.6 (1.6–2.4)	0.186	1.8 (1.8–1.8)	0.18
C-reactive Protein (mg/dL)	3.6 (1.1–7.8)	3.1 (0.8–6.7)	0.421	3.2 (1.0–6.8)	0.505	2.8 (0.5–6.6)	0.344

PSC = primary sclerosing cholangitis; UC = ulcerative colitis; CD = Crohn’s disease, IBD = intestinal bowel disease, IQR = interquartile range.

**Table 4 ijms-24-15431-t004:** Incidence of colorectal carcinoma and hepatobiliary carcinoma in patients with isolated PSC, PSC-IBD and subgroups.

Malignancy	Isolated PSC	PSC-IBD	IBD-Subgroups				
			*p*-Value	UC	*p*-Value	CD	*p*-Value
Malignancy [n %]	7 (17%)	20 (17%) *	0.586	19 (19%)	0.497	1 (7%)	0.305
Hepatobiliary carcinoma [n %]	5 (12%)	11 (9.6%)	0.415	11 (11%)	0.522	0	0.156
Colorectal carcinoma/HGIEN [n %]	0	10 (8.4%)	0.042	9 (9%)	0.041	1 (7%)	0.095

UC = ulcerative colitis; CD = Crohn’s disease, IBD = intestinal bowel disease, HGIEN = high grade intraepithelial neoplasia. * One patient with two carcinomas CRC and hepatobiliary tumor therefore malignancy PSC-IBD n not 21 but 20.

## Data Availability

Because of patient identifying data, original data are not available.

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
