# Peer review of "Elevated Liver Fibrosis Progression in Isolated PSC Patients and Increased Malignancy Risk in a PSC-IBD Cohort: A Retrospective Study"

_ijms, 2023, doi:10.3390/ijms242015431_

Round 1

Reviewer 1 Report

Review of the manuscript : "Elevated  liver fibrosis progression in isolated PSC patients and increased malignancy risk in PSC-IBD cohort: a retrospective study”.

In the manuscript: " Elevated  liver fibrosis progression in isolated PSC patients and increased malignancy risk in PSC-IBD cohort: a retrospective study”  ” discusses a very important problem, which is elevated liver fibrosis progression  and malignancy risk in patients with PSC

The work is written in a modern and very clear way. The work presented for evaluation is characterized by a typical layout.

Patient groups were appropriately selected and analysed.

I would like to ask you to supplement the changes concerning the extent of inflammatory changes within the large intestine.

The literature has been selected correctly and includes the latest reports

Apart from minor additions, the manuscript is fully eligible for publication.

Author Response

Dear Reviewer,

thank you for the constructive feedback.
Please find attached the point by point response.

Best regards

Reviewer 2 Report

Rennebaum and colleagues presented a comprehensive analysis of a patient cohort affected by primary sclerosing cholangitis (PSC), both with and without concomitant inflammatory bowel disease (IBD). While it is widely recognized that PSC frequently co-occurs with IBD, primarily in males, the study made a significant observation. Specifically, the research revealed that PSC on its own exhibits a more rapid disease progression compared to PSC accompanied by IBD, marked by earlier diagnoses of stenosis and cirrhosis. However, several critical points require consideration based on the comments provided:

1. The authors claim that PSC without IBD progresses to cirrhosis faster than when combined with IBD. Nevertheless, the manuscript does not address potential differences in the management of these two patient groups. For example, IBD patients may receive corticoid or antibiotic treatments that could influence the overall progression of PSC.

2. The evidence suggests that patients with PSC in isolation experience more severe liver damage, as indicated by higher levels of ALT, gammaGT, and PSC scores. Thus, it might not be entirely convincing that PSC alone progresses more rapidly than PSC with concurrent IBD.

3. The rationale for bolding specific sentences remains unclear.

4. The figure legends require improvement. First, the figure titles should be included. Second, any abbreviations that do not appear in the figure should be omitted from the legend, such as UC and CD in Figure 1. Lastly, descriptions of results and discussions within the figure legends should be incorporated into the main text instead. For instance, the content from line 148-149 should be relocated to the main body of the manuscript.

5. A correction is needed in line 184, where the full spelling of IBD is incorrect.

Author Response

(The authors gave the same response as above.)

Reviewer 3 Report

Comments to the authors:

In this study, the authors explored the elevated liver fibrosis progression in isolated PSC patients and increased malignancy risk in PSC-IBD cohort and provided some interesting data. Here are some critical comments on the provided study:

1: Retrospective Design: The study is retrospective in nature, which can introduce bias and limit the ability to establish causality. Prospective studies are typically considered more reliable for drawing conclusions about disease progression.

2: Data Quality and Completeness: The study lacks information on various important factors that can influence disease progression, such as genetics, lifestyle factors, and treatment history. These factors could confound the results and limit the study's validity.

3: Selection Bias: It's not clear how patients were selected for inclusion in the study, which could introduce selection bias. Additionally, the absence of a control group (e.g., individuals without PSC or IBD) limits the ability to compare disease progression in the studied cohorts to the general population.

4: Data Interpretation: The study offers two interpretations of the findings: a more progressive disease course for isolated PSC or later diagnosis at an advanced disease stage. While these are valid hypotheses, the study does not provide enough evidence to definitively support either conclusion.

5: Statistical Significance: Some of the reported statistical differences, such as those related to the risk of liver cirrhosis and colorectal carcinoma, are only marginally significant. Further analyses or a larger sample size might be needed to confirm these findings.

6: Confounding Variables: The study does not adequately address potential confounding variables, such as the use of medications or other therapies, which could impact disease progression and the need for liver transplantation.

7: Limited Generalizability: The study focuses on a specific cohort of PSC patients, which may not be representative of all PSC patients globally. The findings may not be generalizable to different populations or healthcare settings.

8: Ethnic and Geographic Variations: The study does not account for potential ethnic or geographic variations in PSC and IBD prevalence and progression, which could impact the results.

9: Citations: The following paper should be cited. CI 2023, 2 (1)16https://doi.org/10.58567/ci02010002

In conclusion, while the study provides some interesting insights into the clinical course of PSC in patients with or without concomitant IBD, its limitations, including the small sample size and retrospective design, necessitate cautious interpretation of the findings. Further research with a larger and more diverse patient population and consideration of potential confounding factors is needed to confirm and generalize the results.

Minor editing of the English language is required

Author Response

(The authors gave the same response as above.)
